# Experimental study on the effect of high-temperature oxidation coal mechanical characteristics

**Xiaoqi Wang**[1,2], **Heng Ma**[1,2], **Xiaohan Qi**[1,2]*, **Ke Gao**[1,2], **Shengnan Li**[1,2], **Xuesong Yang**[1,2]

**1** College of Safety Science and Engineering, Liaoning Technical University, Huludao, Liaoning, PR China,
**2** Key Laboratory of Mine Thermodynamic Disasters and Control of Ministry of Education, Liaoning Technical University, Huludao, Liaoning, PR China

* qxh550799007@163.com

## Abstract

After long-term oxidation and energy storage, broken coal body borehole walls and drainage shaft walls may cause spontaneous combustion during gas extraction. The high-temperature thermal shock caused by the spontaneous combustion of coal incurs thermal damage on adjacent coal, which, in turn, causes changes in the mechanical properties of the coal. However, only a few studies have been conducted in this context, which has limited our understanding of the thermal damage characteristics of coal bodies in such situations. This study aimed to experimentally investigate the correlation between the crack evolution law and the mechanical properties of coal bodies at different temperatures (50–300°C) using heat-force loading considering Ping Mei No. 10 coal mine as the research object. The results suggest that the coal body experiences a large amount of visible damage, and becomes increasingly complex. At 50–300°C, some indexes (such as longitudinal wave velocity, Poisson's ratio, compressive strength, elastic modulus, impact energy index, and pre-peak strain) are positively correlated with temperature. In addition, the dynamic failure time and temperature show a negative correlation, and the overall change slope is small. The relationship between each index and temperature at 200–300°C is opposite to that at 50–200°C, and the overall change slope is larger. Moreover, when the oxidation temperature exceeds 200°C, the destruction of the coal body changes from elastic brittleness to ductility-plasticity. High-temperature oxidation incurs irreversible thermal damage of coal. Hence, it is necessary to focus on the changes in mechanical properties of coal after a spontaneous combustion process is extinguished.

## 1 Introduction

Because external resources of coal mines are increasingly being depleted, and the state of deep mine mining is gradually shifting, coal seam storage environments are presenting new characteristics [1]. Pressure relief and drainage of a coal seam can reduce gas outbursts as well as compound disasters involving gas outbursts and spontaneous coal combustion. The air leakage

**Data Availability Statement:** All relevant data are within in the paper.

**Funding:** This project was funded by the Natural Science Foundation of China (No. 52074148).

**Competing interests:** The authors have declared that no competing interests exist.

intensity increases with increasing cracks in the coal, which leads to the oxidation of coal. Eventually, spontaneous combustion of coal occurs. Fig 1 shows that the protective layer relieves the pressure of the protected layer, leading to the further development of fissures and elimination of gas outbursts in the protected layer area. The coal is oxidized and the spontaneous combustion is induced when oxygen enters the protective layer through the cracks. Fig 2 shows that in the pores and cracks of coal, the mechanical characteristics decrease during the high-temperature increase. The coal is further destroyed because coal in a deep mine is in a high in situ stress environment [2, 3]. Some problems occur owing to the velocity of evolution of cracks, and the density of cracks increases sharply. On the one hand, the extraction rotary hole that is needed in the process of gas drainage is destroyed. The adsorbed gas in the pores is connected to the outside through the main crack channel [4]. These two aspects will make the fire more serious. The broken coal located around an extraction rotary hole undergoes a long-term oxidation process, leading to coal body energy storage and spontaneous combustion. The high temperature generated by combustion causes thermal damage to the coal near the extraction rotary hole, which causes changes in the mechanical properties of coal. Some problems should be addressed in the study of deep mine coal mechanics [5, 6]. At the beginning of 2013, spontaneous combustion was discovered in the No. 10 coal mine during the process of gas drainage and drilling in the 15–24080 working face. According to statistics, 30 boreholes were on fire, and infrared temperature measurements at the fire site showed that the temperature reached 230°C. The high-temperature oxidation effect results in changes in the physical and mechanical properties of coal, which affects its stability. The effect does not disappear completely when the flame is extinguished. If the coal seam is evaluated before high-temperature oxidation, it will inevitably cause errors in the evaluation results and become a new safety hazard in the mine.

Temperature has a significant influence on the mechanical properties of coal. Scholars at home and abroad have carried out a series of studies on the influence of temperature on rock damage and mechanical properties.

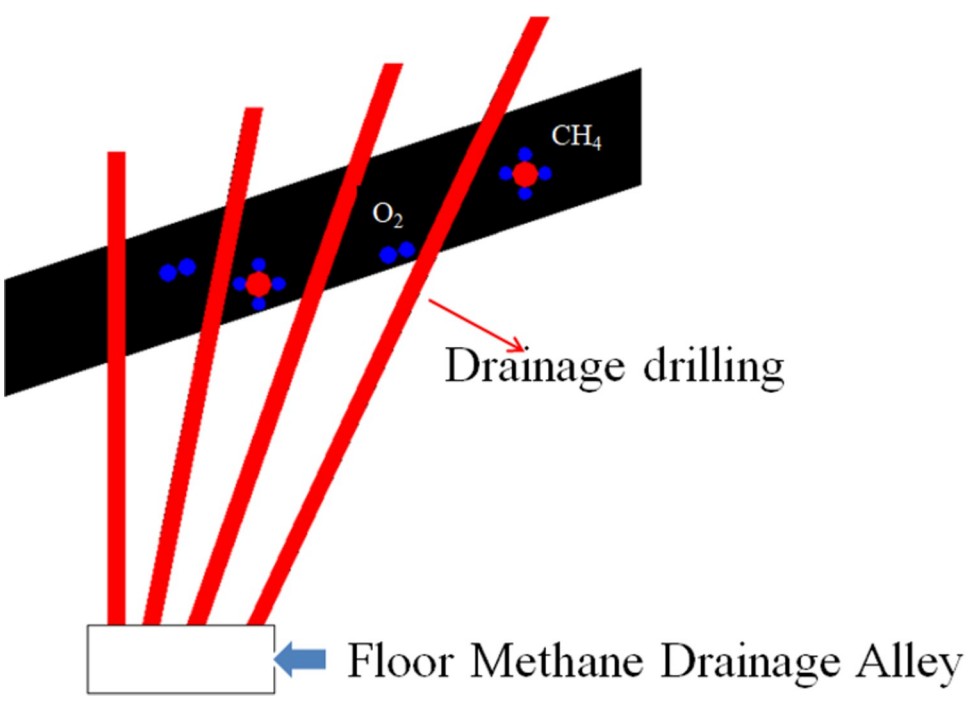

**Fig 1. Protection layer mining.**

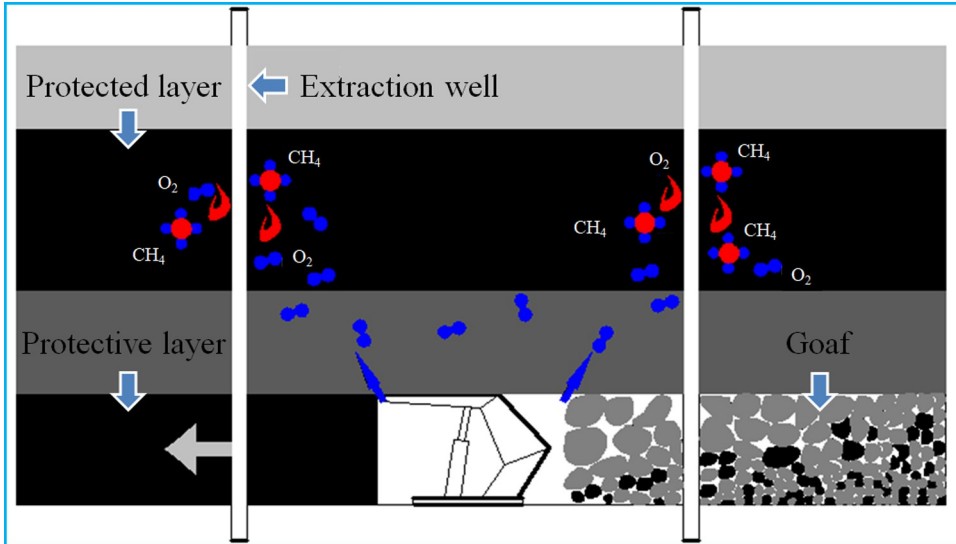

**Fig 2. Floor gas drainage.**

Gan et al. [7] analyzed the molecular structure of coal and studied the influence of temperature on the coal mechanical properties and, its microstructure, and the reasons for the evolution of coal mechanical strength. Tao et al. [8] performed fractal dimension analysis and infrared spectroscopy analysis on coal samples treated at different temperatures, and obtained the formation process of pores and cracks during the pyrolysis process and the changes in microphysical parameters with increasing temperature. Lan et al. [9] analyzed the mechanism of microcracks inside and on the surface of high-temperature treated coal samples through numerical simulation and experimental research. Song et al. [10] studied the heating rate and macroscopic spontaneous combustion characteristics of different coal particle sizes during high-temperature oxidation. Jiang et al. [11] used thermo-gravimetric analyzers and a down reactor and, discussed the influence of temperature and particle size on the rapid pyrolysis behavior of coal at high temperatures.

Yang et al. [12] used CT scanning electron microscopy (SEM) to study quantitatively the evolution of coal fractures during heating from the macro and micro perspectives, and expounded the influence of thermal damage on the mechanical properties of coal and the development of pores and fissures. Yavuz et al. [13] gradually heated the rock to specific temperature levels of 100, 200, 300, 400, and 500˚C, and gradually cooled it to room temperature without causing thermal shock to study the effect of thermal damage on the micro scale and the influence of structure, bulk density and effective porosity.

Ultrasonic technology can reflect the mechanical properties of coal. Hassani et al. [14] used the ultrasonic technology to study the changes in internal defects of loaded coal quantitatively. Hong et al. [15] used microwaves to heat a coal sample, and utilized camera photos and ultrasonic methods to evaluate the crack growth on the surface and inside of the coal. The results show that as the heating time increases, the fracture caused by microwave energy will increase significantly, and the fracture network tends to become more complicated. Liu et al. [16] used the ultrasonic technology to study the changes in internal defects of coal after loading in a quantitative manner, and provided a theoretical basis for applying this technology to determine the structural stability of coal and predict disasters related to coal or rock dynamics. Morcote et al. [17–20] used ultrasonic testing to evaluate the influence of thermal effects on

wave velocity and P-wave modulus. The effectiveness of the ultrasonic evaluation method was confirmed. The factors that affect the speed of ultrasonic propagation in coal (such as coal rank, water saturation, porosity, and permeability) were studied, and the results showed that ultrasonic speed can be used to evaluate the degree of fracturing of coal.

Huang et al. [19, 21] studied the effects of thermal shock on the compressive strength, elastic modulus and wave speed of deep rocks after they were subjected to different temperatures, and discussed the rock deformation and fracture mechanism after rock materials were subjected to a high-temperature treatment. Wu et al. [22] heated sandstone at a temperature of 20–1200˚C, studied the changes in shape, volume, weight, and density of the sandstone before and after high-temperature treatment, along with the speed of transverse and longitudinal elastic waves passing through the test sample, and analyzed the stress-strain response. Uniaxial compressive strength, Elastic modulus and Poisson's ratio. Ma et al. [5] completed an experimental study on the influence of temperature on the mechanical properties of coal, obtained the stress-strain relationship of coal at different temperatures, and clarified the influence of temperature on its compressive strength and elastic modulus. Pan et al. [23] studied the changes in the mechanical properties of combustible coals after oxidation, and oxidized the coal samples through programmed temperature rise, and then proposed changes in the mechanical parameters of different oxidized coals. Zhang et al. [24] explored the influence of temperature on the mechanical strength of granite by carrying out real-time high temperature compression experiments on granite samples, clarified the difference in physical and mechanical properties of granite under real-time high temperature and thermal shock conditions, and revealed that thermal shock damage is broken the Rock mechanism. Yu et al. [25] revealed the different changing laws of the uniaxial compressive strength of tight sandstone at and after high temperature. Zhang et al. [26] studied the effect of high temperature on the strength and elastic modulus of soft rock. Liu et al. [27] studied the longitudinal wave velocity, compressive strength, failure form and damage characteristics of marble measured at different temperatures through uniaxial compression tests. Cha et al. conducted experimental research on the mechanical section steel of coal measure mudstone that experienced different temperatures, and discussed the influence of temperature on peak strength [28], elastic modulus, deformation modulus and Poisson's ratio.

In recent years, the physical and mechanical properties of coal have been investigated in China and abroad. Efforts to characteristic the influence of temperature on the mechanical properties of coal and the quantitative description of the meso damage of deep coal have been studied. However, there are few reports on the quantitative description of the microscopic damage of high-temperature oxidized coal and research on the physical and mechanical properties of coal. In this study, by quantitatively describing the thermal damage of oxidized coal at different temperatures, the correlation between thermal damage and mechanical properties of high-temperature oxidized coal was explored. As a result, a reference for the stability evaluation of high-temperature oxidized coal, together with a reference for the prevention and control of the instability of ignited coal in coal drilling and extraction wells.

## 2 Thermal-stress loading test

### 2.1 Coal sample selection

Auxiliary measuring equipment was used to carry out thermal stress loading tests on high-temperature oxidized coal. The coal sample was taken from the 24130 working face of the Ping Mei No. 10 mine in Henan province, China, and mined 1073–1173 m deep coal seam. The coal sample was composed of 1/3 coking coal and coking coal. Table 1, presents the coal seam industrial—analysis results, and the parameters of the test device.

**Table 1. Coal sample industrial analysis results (%).**

| Coal sample | Moisture $M_{ad}$ | Ash $A_{ad}$ | Volatile matter $V_{ad}$ | Fixed carbon $F_{ad}$ | Calcite | Pyrite | Clay mine |
|---|---|---|---|---|---|---|---|
| Fat coal | 1.37 | 18.86 | 23.27 | 43.09 | 2.31 | 0.60 | 10.50 |

## 2.2 Preparation of coal sample

The coal sample was cylindrical, with a size of φ 50 × L100 mm, and the parallelism of the upper and lower ends of the cylinder, flatness, and diameter at both ends of the test piece were less than 0.02, 0.50, and 0.20 mm, respectively. Meanwhile, the coal samples were numbered MY1-MY53. The primary fracture structure of coal has a great influence on its mechanical properties, which leads to many variable factors in the research process, and the results obtained are often highly discrete. Thus, the selected coal samples should be taken from the same coal as much as possible, and the coal sample bedding direction should be consistent in the selection process. A non-metallic ultrasonic detector was used for the ultrasonic testing of 53 coal samples, and each specimen was tested three times. The error caused by the test results was reduced to pass coal samples with similar properties. The 30 selected coal samples were divided into six groups according to temperature, and each group maintained three valid specimens. (The oxidation treatment of the specimen was performed at different temperatures (T = 50, 100, 150, 200, 250, and 300˚C). When there was a large error, the number of experiments was supplemented appropriately. The number of specimens used is listed in Table 2.

## 2.3 Test method

Fig 3 shows the experimental flow chart. A uniaxial compression test was carried out on the coal samples after oxidation treatment at different temperatures, and the oxidation temperatures were set to 50, 100, 150, 200, 250, and 300˚C. The coal block was ignited at 320–380˚C, and the highest oxidation temperature was set to 300˚C to prevent the coal sample from burning. In addition, taking 50˚C as the normal temperature of the mine, the temperature of the deep coal seam at the sampling location was approximately 50˚C. All the coal samples were oxidized and stored in the same environment during the test. After a coal sample was cooled to room temperature, it was placed in a thermostat for storage. The coal sample was loaded using the machine displacement control method, and the loading rate was 0.1mm/min. During the loading process, scanning was performed according to the displacement loading value on the software interface of the microcomputer servo instrument, and the sample was scanned with a three-dimensional scanning system every 1.0 mm to obtain a point cloud image of the coal specimen. The detection time of the nonmetal ultrasonic detector was the same as the three-dimensional scanning time, and the ultrasonic signal parameters when the coal sample was loaded with the same displacement during the experiment were collected. In addition, a comparative coal sample was prepared in advance, and loaded without force. The microscopic damage of the coal sample was measured and imaged with a 4 K scientific camera after oxidation and cooling at different temperatures. The evolution process of thermal damage fracture of high-temperature oxidized coal occurs through an image mosaic. An HC-U7 nonmetal

**Table 2. Experimental conditions and number of samples.**

| Temperature /˚C | 50 | 100 | 150 | 200 | 250 | 300 |
|---|---|---|---|---|---|---|
| Design test / number | 3 | 3 | 3 | 3 | 3 | 3 |
| Final test / number | 6 | 8 | 5 | 5 | 3 | 3 |

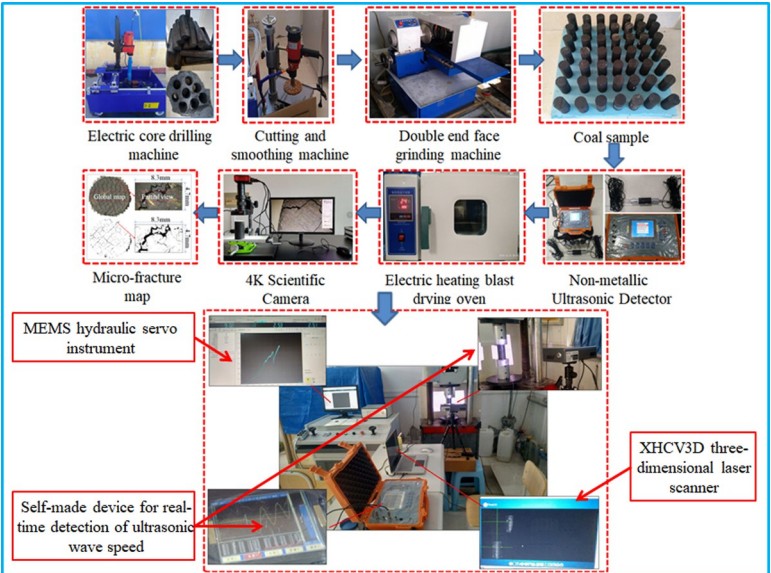

**Fig 3. Experiment flow chart.**

ultrasonic detector was used to obtain the coal sample waveform curve, wave velocity, and wave amplitude. The three-dimensional scanning system (XHCV3D) scans different loading displacements to obtain a three-dimensional point cloud during the coal sample loading process. The target paper on the surface of the coal sample was measured by Geo-magic studio post-processing software, the transverse and longitudinal strains were obtained, and the Poisson's ratio was calculated.

# 3 Analysis of test result

## 3.1 Physical and mechanical properties of oxidized coal

**3.1.1 Physical parameter analysis.** The sound wave, sound velocity and wave amplitude were detected based on the HC-U7 non-metallic ultrasonic detector. Wave velocity is an indirect characteristic of coal damage. Formula 1 can be used to calculate the coal damage factor ($D$) at different temperature [29].

$$D = 1 - \left(\frac{V_{PT}}{V_P}\right)^2 \tag{1}$$

Where, $V_P$ is the untreated longitudinal wave velocity of the coal sample (m/s), and $V_{PT}$ is the longitudinal wave velocity of the coal sample processed at different temperatures (m/s).

The structure of coal is very dense, and is a heterogeneous natural material composed of crystalline particles and pores. Non-uniform deformation stress and thermal cracking are produced with increasing temperature, owing to the different coefficients of thermal expansion. Because coal is subject to long-term geological activities, its structure and mineral composition is complex. Two micro-element bodies in the coal body were selected for analysis, and it was assumed that these two micro-element bodies were constrained. The mechanics of coal micro-element were expressed in three ways. First, the two different types of particle, are closely adjacent to each other. Second, the coefficients of thermal expansion and modulus of elasticity of the two types of substances are $\lambda_1$, $\lambda_2$ and $E_1$, $E_2$ respectively. Third, the temperature increased

from room temperature. Then, the thermal stress was calculated as follows [30].

$$\Delta\sigma = (\lambda_1 + \lambda_2)\Delta T \frac{E_1 E_2}{E_1 - E_2} \qquad (2)$$

Cracks occur around the crystal particles when the ultimate strength of $\Delta\sigma$ is exceeded $\sigma$. The change in $\Delta\sigma$ is determined mainly by $\Delta T$. Compressive stress is generated between the crystal particles during heating, and the internal composition and structure will cause complex physical or chemical changes. All of the above will lead to significant changes in the mechanical properties of coal, resulting in thermal cracking. The mechanical properties of coal include parameters such as density, porosity, water content, deformability, and compressive strength.

The test results showed that the coal quality and coal density tended to decrease as the temperature increased. During the test, it was determined that the desorbed gas released from coal mainly contained $H_2O$, $CH_4$, $CO_2$, $C_2H_6$, $N_2$, and $O_2$. When the temperature was 50–200˚C, the moisture escaped and a small amount of attached gas was discharged. Therefore, the space originally occupied by water and gas was released and connected. When the temperature was 50–200˚C, the moisture escaped and a small amount of attached gas was discharged. Therefore, the space originally occupied by water and gas was released and connected. At the same time, the matrix in the coal, especially some organic components, became soft under the action of high-temperature oxidation, which made the pores squeeze and deform, thus affecting the pore structure. At this stage, what happens in coal is a physical change. The overall pore structure parameters of coal showed little change, but there were still small fluctuations. When the temperature exceeded 200˚C, the amount of discharged water decreased, and the concentrations of $CH_4$, $CO_2$, and other gases gradually increased. During the heating and oxidation process, a burning pungent smell was noticed, indicating that the organic matter and minerals in the coal had undergone a pyrolysis reaction, and the loss of coal moisture and pyrolysis gas resulted in a gradual decline in coal quality. At this stage, physical and chemical changes occurred inside the coal, and the coal was decomposed by the high temperature and a large amount of gas was produced. These gases continued to be heated, and their volume expanded sharply, resulting in large local tensile stresses in the coal. In addition, the volume of pores could increase, and the pore walls with weaker structures could be broken. Many large-scale interconnected structures were formed owing to the penetration between the pores. As a result, the number of pores and, average pore diameter increased. In addition, as the temperature increases, there was a significant influence between the pores and the coal skeleton. The sound velocity test of coal samples treated at different temperatures by non-metallic ultrasonic waves can be used to evaluate the thermally damaged coal samples quantitatively. Fig 4 shows the wave velocity-temperature curve. After the coal body was processed at different temperatures, the change rates of the coal sound velocity were -0.36%, -14.59%, -19.22%, -24.56%, -37.72%, and -45.91%, respectively. As shown in Fig 4, as the operating temperature increased, the tightness of coal decreased, and the pores increased. This caused the coal sound velocity to decrease, and the coal damage factors were 0.007, 0.271, 0.347, 0.431, 0.612, and 0.707, respectively.

This indicates that the degree of coal sample damage becomes more serious as the temperature increases. The structure of coal and the properties of the material change significantly under the action of high temperature. During the propagation of ultrasonic waves, the further development of internal cracks will cause refraction and diffraction, which increases the travelling resistance of elastic waves as the temperature rises further. This prolongs the propagation time of the sound waves in the coal sample, resulting in a reduction in the wave velocity and attenuation of energy. Owing to the high-temperature oxidation-cooling to room temperature

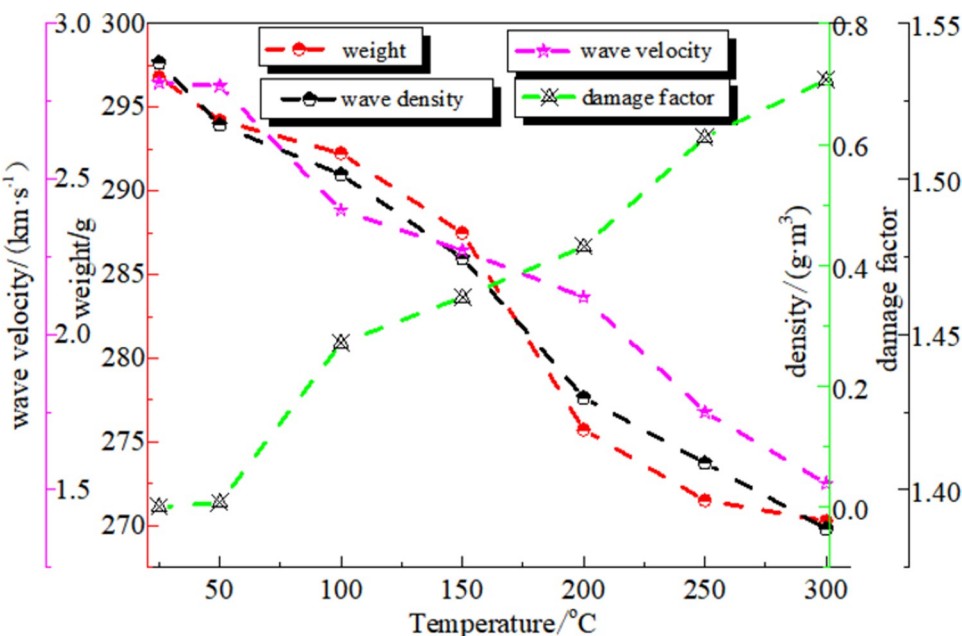

**Fig 4. Relationship between coal quality, density, sound velocity, damage factor and temperature.**

treatment, the damage of the coal is irrecoverable. Therefore, after the spontaneous combustion of the coal seam is extinguished, it is still necessary to pay attention to the changes in mechanical properties caused by the thermal damage of the coal seam.

**3.1.2 Image quantitative analysis under the meso damage of coal.** Fig 5 shows a micro-level thermal damage image obtained by photographing coal samples at different temperatures using an electron microscope camera. Owing to the limitation of the field of view, the 4 K scientific research camera can only observe a small area on the surface of the test piece. The developing size is 8.3 mm × 4.7 mm. The solution to the problem is to move the microscope along the observation surface to take continuous pictures, and then stitch the images together in the order the pictures are taken to obtain a larger range of images. To better observe the cracks macroscopically, the brightness and color of the full image of the surface cracks of the spliced coal sample after different temperature treatments were adjusted, as shown in Fig 6.

As shown in Fig 7, coal is cemented by coal crystal particles with large densities, and it has an irregular spatial structure. Coal is dense at 50˚C, and there are very few micro-cracks. A clear and tortuous line can be observed on the coal surface. The thermal fracture of coal

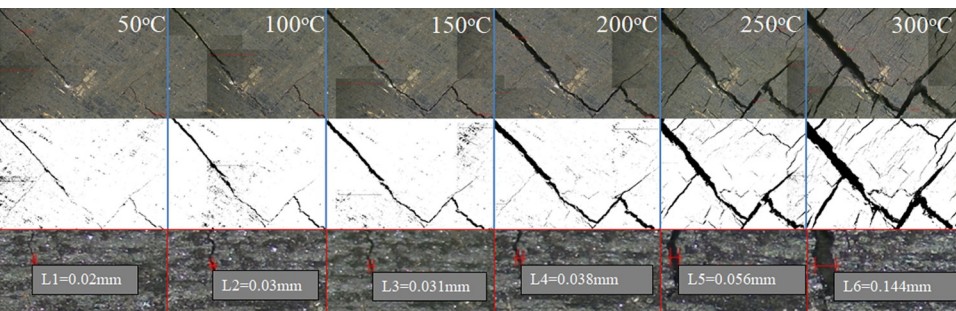

**Fig 5. Microscopic image of coal thermal damage after oxidation at different temperatures.**

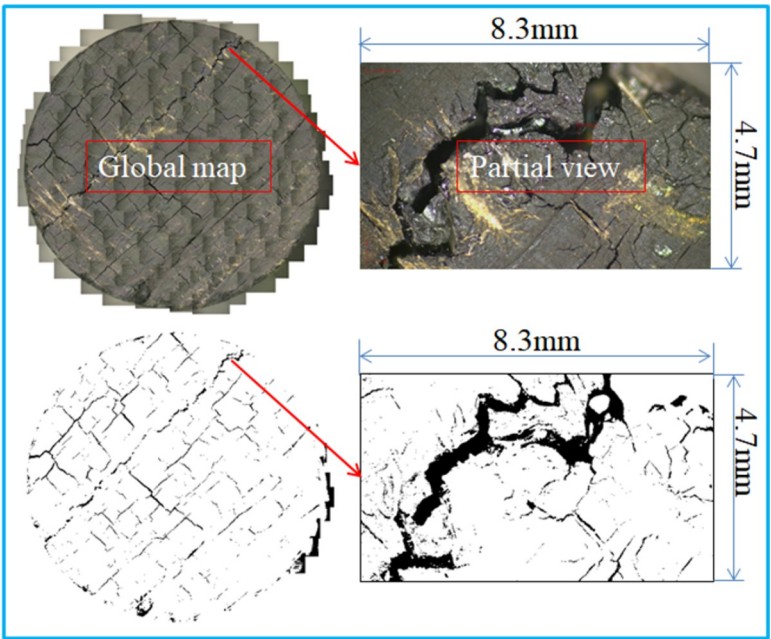

**Fig 6. Microphotograph of the end face of the oxidized coal sample and crack extraction.**

gradually evolves and develops with an increase in the oxidation temperature. When the temperature is 50–200˚C, the first thermal fracture area is a tortuous crack on the original coal surface. At this stage, crack propagation becomes the primary crack growth. There are few microcracks at the grain boundary, and the number of new cracks is small. The crack width develops from the original 20 um to 36 um at 200˚C. When the temperature is 200–300˚C, the width increases from 36 um to 144 um, and the range of increased is as high as 300%. There are more cracks around the coal crystal particles, and in addition to the formation of H-shaped through cracks, most of them have weak links. The length of the thermal crack increased more than before, and its width also expanded from the original 36 um to 56 um. At 300˚C, the micro-

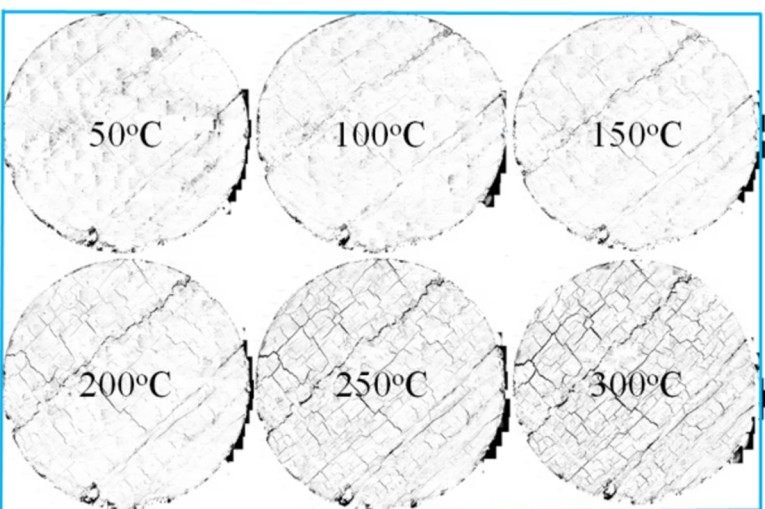

**Fig 7. Extraction map of oxidized coal fracture network.**

cracks further expand and penetrate, the entire coal sample. The thermal damage causes the surface of the coal to form a three-dimensional irregular fracture network structure.

## 3.2 Full stress-strain curve

The information on damage evolution of the coal samples can be obtained from their stress-strain curves after heat treatment at different temperatures. The stress-strain curve obtained after the high-temperature treatment of coal at 50, 100, 150, 200, 250 and 300°C comprises five stages: the fracture compaction stage of primary pores, linear elastic stage, plastic weakening, weak surface damage failure, and full failure. The details are as follows.

Fracture compaction stage of the primary pores. The compactness gradually improves at this stage. The amount of axial strain in the compaction stage reflects the degree of coal damage caused by the temperature of the coal sample. Because the coal medium is dense, the strain in the compaction stage of the coal sample is small after the external load is applied when the processing temperature is low. Under the same strain, the higher the treatment temperature, the greater is the stress value.

In the linear elastic stage, the axial stress causes the coal sample to accumulate elastic potential energy quickly, and the stress increases linearly with the increase in deformation. The unit strain stress increases faster in coal samples under normal temperature.

During the process of plastic weakening, the coal samples at 50–150°C show good linear characteristics, and the changing trend of the stress-strain relationship between 200–300°C is more obvious than the changes in coal samples at 50–150°C. As the axial compression continues, the magnitude of the stress change decreases, and its change trend deviates from linearity. New and existing cracks continue to grow at this stage. These cracks are connected, and the deformation of the coal body structure is irreversible.

Before reaching the peak, some curves have a descending stage, and the axial pressure exceeds the adhesion and friction of the bedding surface during weak surface damage failure. There is a phenomenon in which shear slip occurs inside the coal body, and it is accompanied by energy release. Near the peak, some coal bodies have a significant "multi-peak effect". The concrete manifestation is the closure, expansion, and re-closure of the joint fissures. Simultaneously, a rock bridge exists between the joints. When the rock bridge penetrates, the stress-strain curve shows a peak.

In the full destruction stage, the stress reaches the ultimate bearing capacity of the coal body after the peak. Furthermore, the elastic energy accumulated in the coal is quickly released and destroys it, as shown in Figs 8–13. The internal fractures of the coal develop and form macroscopic cracks, and the coal loses its bearing capacity as the deformation increases. Owing to the increase in coal cracks and fissures, the compressive strength of the coal itself is reduced. According to the stress-strain curves of coal samples at different temperatures, it is concluded that different temperatures will have a noticeable impact on the coal failure form after the peak value. Between 50 and 150°C, there is a sharp drop in the curve after the peak intensity, the coal sample is destroyed rapidly, and the fractures are concentrated. The stress of the coal sample at 50°C decreases linearly and then increases linearly many times. The crack surface of the coal sample is broken, and the angle of the crack surface causes the coal sample to exhibit shear slip.

As the degree of oxidation increases, the peak intensity gradually decreases. The peak strain expands gradually during the compaction stage. However, the slope (elastic modulus) of the linear elastic phase gradually decreases, the step after the peak decreases significantly, and the residual strength increases.

The strain variable can be used as the basis for judging the plasticity of the coal body from the time the coal receives the load to the peak compressive strength. As shown in Fig 14, the

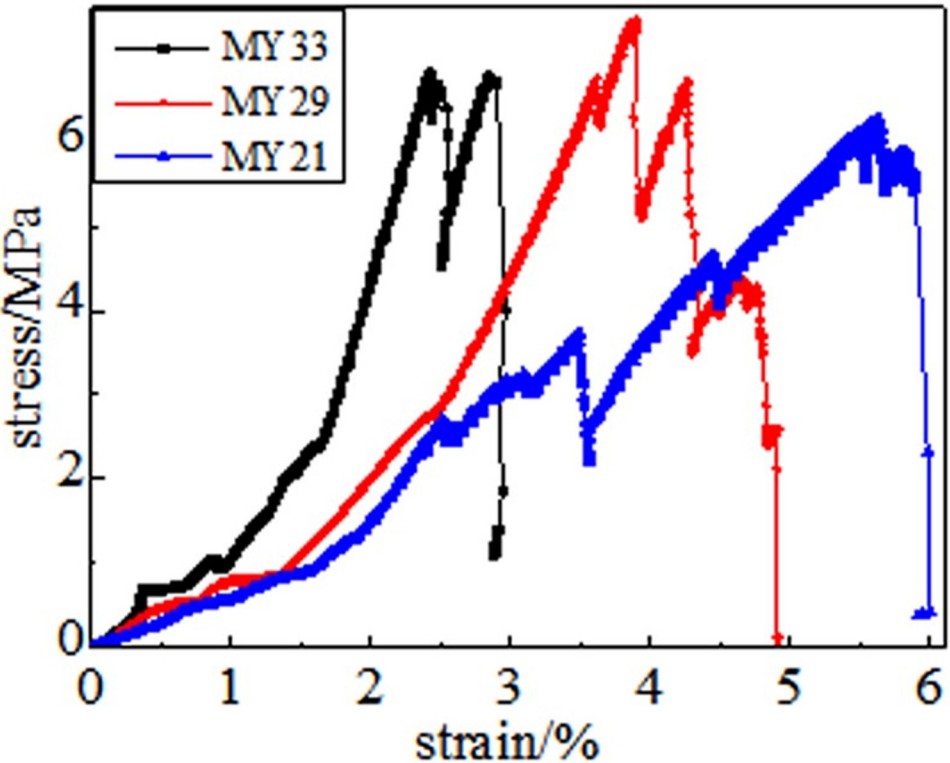

**Fig 8. 50˚C.**

amount of strain when the peak strength is reached at 50, 100, 150, 200, 250, and 300˚C is 3.91%, 4.02%, 4.26%, 4.35%, 4.39%, and 4.97%, respectively. This shows that as the temperature increases, the internal damage of the coal sample increases at the same strain rate. The

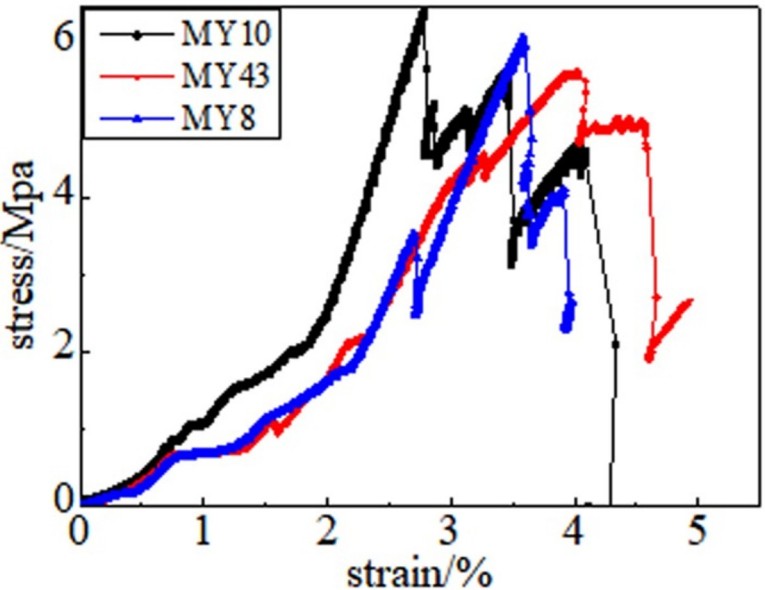

**Fig 9. 100˚C.**

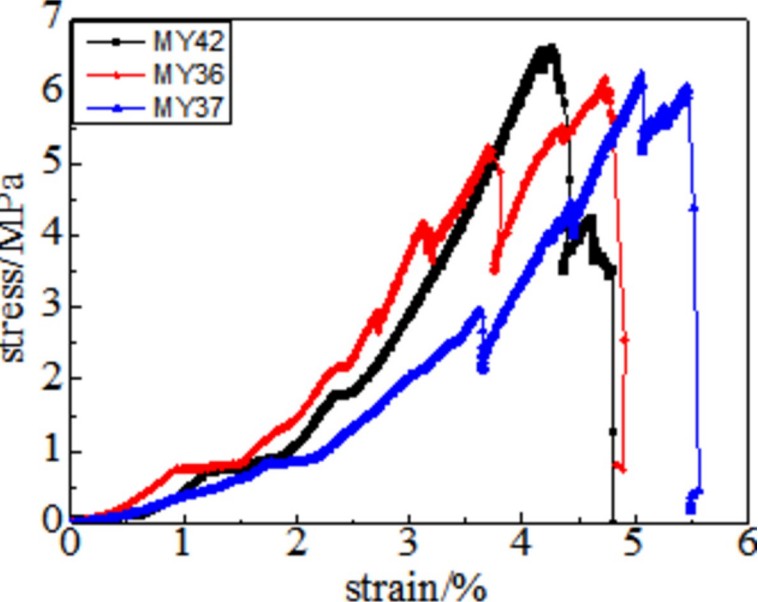

**Fig 10. 150˚C.**

internal cracks are closed first, and the higher the initial damage, the longer is the compaction stage. The internal cracks strengthen the coal sample plasticity when the coal sample is subjected to an external action. This is because the high temperature causes the coal sample to oxidize, which severely changes the structure of coal pores and fractures and the degree of cementation between coal particles. The integrity of the coal sample is reduced, which is characteristic by plasticity. It can be concluded that the high temperature strengthens the plasticity

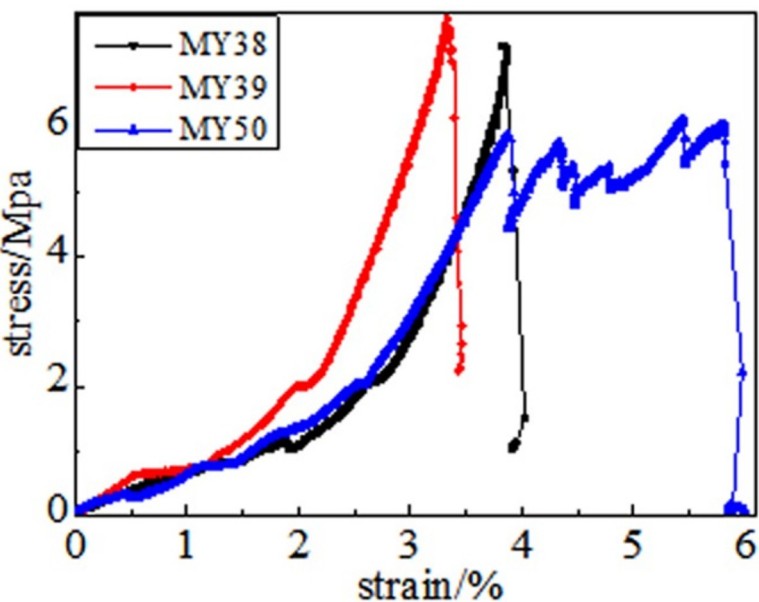

**Fig 11. 200˚C.**

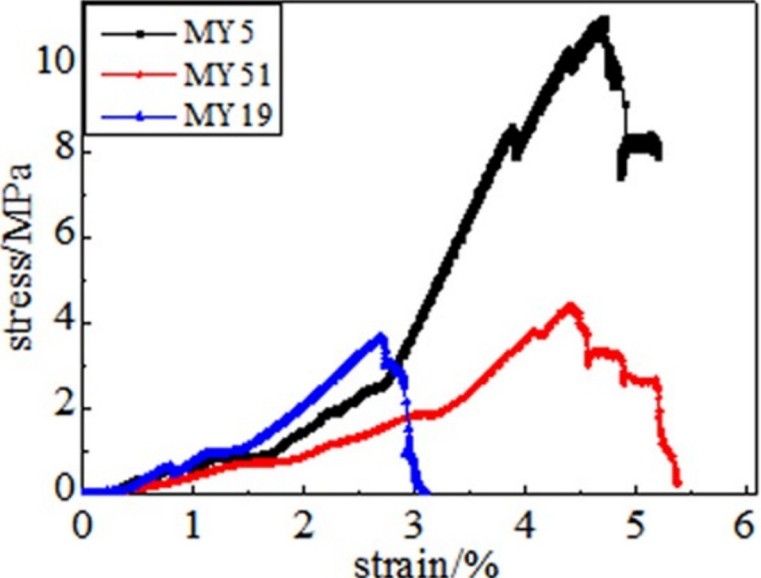

**Fig 12. 250˚C.**

of the coal sample and weakens its mechanical properties, and the failure mode of the coal sample changes from brittleness to plasticity.

It can be observed from the curve in Fig 14 that the mechanical properties of coal are different, its fracture after peak strength is unstable, and the form of expansion is also different. When the temperature reaches 50–100˚C, after the peak strength, the stress will drop in a "step-like" manner. The stress manifests itself as a linear rise to a fixed value, then it falls again, and after it is repeated three times, it loses resistance when it falls after the first fall. When the temperature is in the range of 150–200˚C, the stress-strain curve of the coal sample shows a

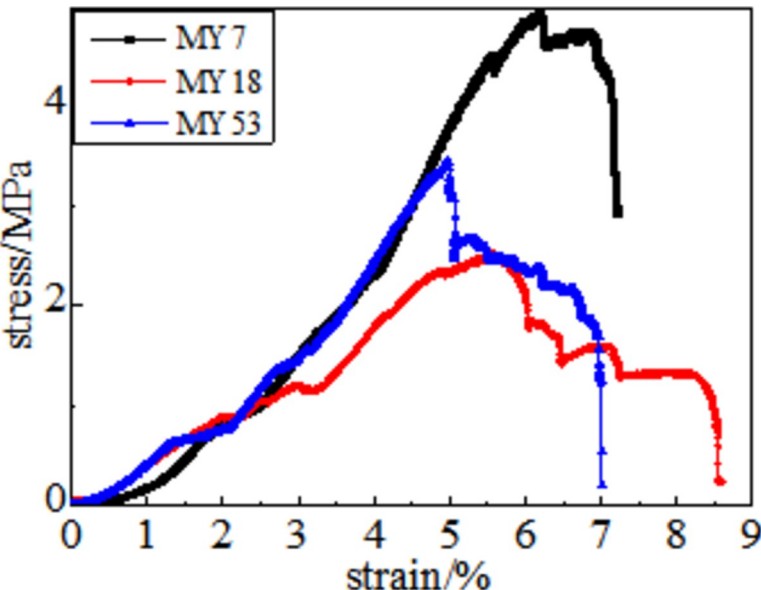

**Fig 13. 300˚C.**

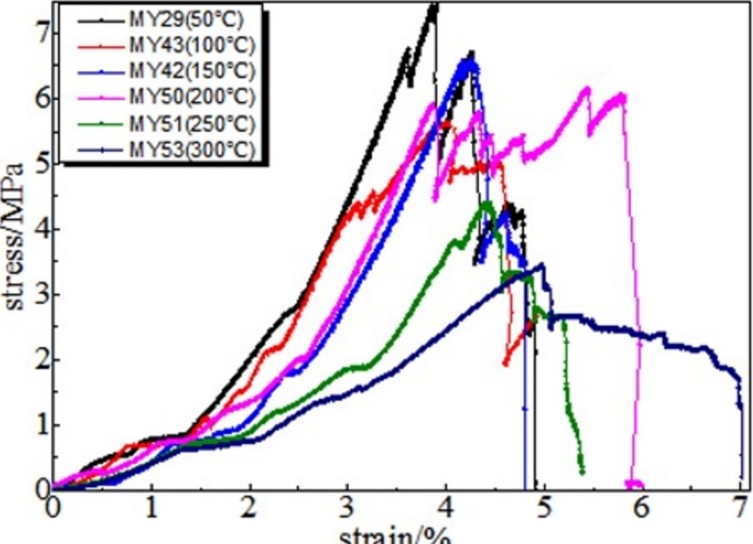

**Fig 14. Typical coal sample stress-strain curve.**

stress slope drop, followed by two visible linear drops. However, the ductility of the coal samples at 250–300˚C is significantly enhanced, and the post-peak stress decreases in a gentle slope without a sudden drop. The residual strength and ductility are enhanced, and the phenomenon of stress drop gradually weakens or disappears.

## 3.3 Influence of coal oxidation on the mechanical properties of coal

The stress-strain curve of coal and the macroscopic failure characteristics of the coal samples were selected for the analysis. The three coal samples exhibited very close compressive strengths and similar failure characteristics and were selected for analysis in each group of coal samples. The curves of compressive strength, elastic modulus, and Poisson's ratio with temperature are shown in Fig 6.

After the components in the coal are subjected to thermal shock, the internal heat energy gradually dissipates. In addition, the high temperature causes the internal moisture to evaporate quickly and decompose mineral impurities, and the porosity increases. At the same time, the uneven expansion of different components in the coal during the heating process and the residual stress generated after cooling caused the cracks to develop rapidly. Owing to the pyrolysis effect, more damage is caused. This is the main reason why the strength of coal generally weakens with an increase in the thermal shock temperature.

As shown in Fig 15, after the coal is oxidized at 50˚C higher than the deep well ambient temperature, the compressive strength generally decreases with temperature. After oxidation from 50–200˚C, the uniaxial compressive strength does not change much with the increase of temperature; the average peak strength of coal at 200–300˚C decreases rapidly, and the average peak stress decreases from 8.13MPa at 200˚C to 250˚C 6.4MPa at 300˚C and 4.0MPa at 300˚C, the reductions reached 21.28% and 50.80%. Compared with the uniaxial compressive strength of 9.92MPa at 50˚C, the average decrease of the compressive strength at 250~300˚C is 35.48% ~59.68%. The porosity, water content, internal structure, density, original cracks, and pores change after the temperature treatment. These factors determine the compressive strength of coal. It appears that microstructural changes will have an impact on its mechanical properties after the coal is processed at different temperatures. At the same time, the inhomogeneous

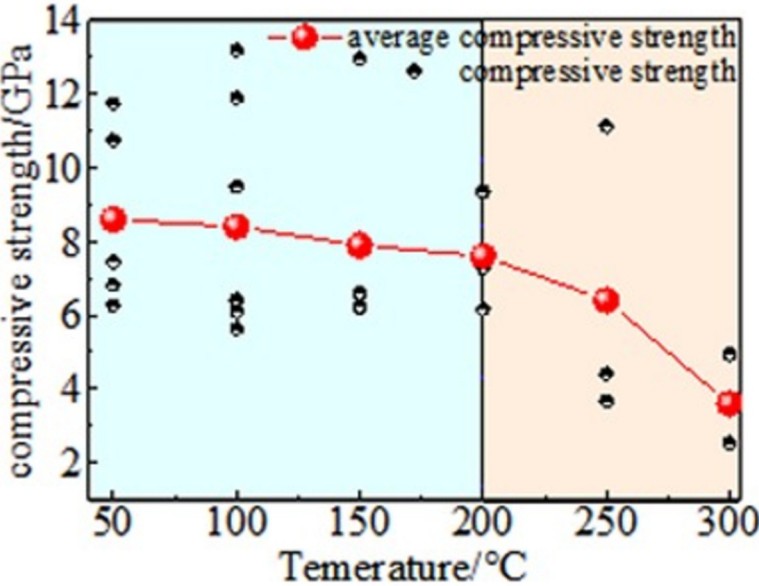

**Fig 15. Compressive strength.**

expansion in the composition and residual stress generated after cooling make the cracks develop rapidly during the heating process. This severely reduces the degree of cementation between the pore and fracture structure of the coal and coal particles, and reduces the integrity of the coal sample. With the gradual increase in the treatment temperature, the load-bearing capacity of the coal skeleton exhibits the characteristic of continuous decrease. The deformation of coal increases owing to the crack propagation. When subjected to axial loading, "airfoil-tensional" fractures appear around the original fractures, and the surrounding pores are connected. The cement surface between particles is prone to rupture, resulting in particle slipping, swelling, and fragmentation. With an increase in the coal processing temperature, the coal sample strength softening coefficient and deformation parameter reduction coefficient decreases as a whole.

It can be observed that after the thermal fracture of the coal sample, the porosity increases, and the internal structure becomes complicated. The mechanical strength of the coal is seriously damaged. The state of stress concentration near the circular hole in the case of unidirectional uniform stretching is analyzed to express the state of stress concentration around the pore according to the theory of elasticity. The stress component on the plate is expressed as follows [31].

$$\begin{cases} \sigma_r = \dfrac{\sigma}{2}\left(1 - \dfrac{a^2}{r^2}\right) + \dfrac{\sigma}{2}\left(1 - \dfrac{4a^2}{r^2} + \dfrac{3a^4}{r^4}\right)cos2\theta \\[2mm] \sigma_\theta = \dfrac{\sigma}{2}\left(1 + \dfrac{a^2}{r^2}\right) - \dfrac{\sigma}{2}\left(1 + \dfrac{3a^4}{r^4}\right)cos2\theta \\[2mm] \tau_{r\theta} = -\dfrac{\sigma}{2}\left(1 + \dfrac{2a^2}{r^2} - \dfrac{3a^4}{r^4}\right)sin2\theta \end{cases} \quad (3)$$

As shown in Formula (3), the cross section of the hole edge ($r = a$) is perpendicular to the stretching direction. Owing to the hoop stress ($\sigma_{\theta max} = 3\sigma$), it decays rapidly at a distance from the periphery of the hole. If the number of pores in the coal body is large, high-stress areas are

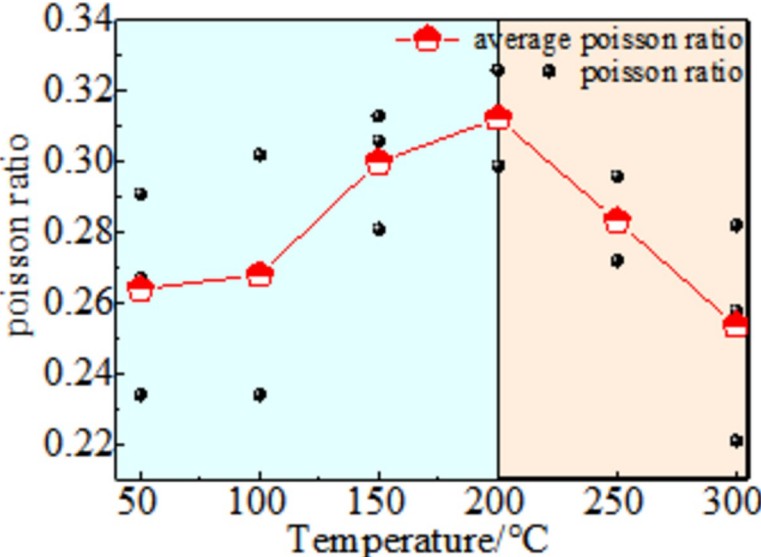

**Fig 16. Poisson ratio.**

easily formed around the small pores, which will cause stress concentration and reduce the macro-mechanical strength of coal.

Poisson's ratio is the absolute value of the ratio of hoop strain to axial strain during uniaxial compression of coal, which is an important parameter for studying the coal deformation characteristics. The relationship curve of Poisson's ratio with temperature after coal thermal shock is shown in Fig 16. The change in the Poisson's ratio of coal with the thermal shock temperature is divided into two stages. The elastic modulus shows a gentle upward trend with the increase in temperature, but the change range is not large when the temperature is 50–200˚C. Owing to the different expansion rates of the particles and overall expansion state, the sample is bound by the axial stress, the axial deformation changes little, the hoop deformation is relatively large, and the Poisson's ratio of the coal sample exhibits an upward trend at this stage. At 200–300˚C, the Poisson's ratio of the coal sample decreases with an increase in temperature. Because the coal body is destroyed in this stage, the mechanical strength decreases rapidly, and the axial strain increases rapidly, resulting in a decrease in the Poisson's ratio.

The elastic modulus is an important reference index for mechanical materials. The definition is as follows [32].

$$E = \frac{\Delta\sigma}{\Delta\varepsilon} \qquad (4)$$

where, $\Delta\sigma$ and $\Delta\varepsilon$ are the axial stress difference and strain difference between the end point and the starting point, respectively during the elastic stage.

Fig 17 shows that 200˚C is the threshold temperature for the change of coal elastic modulus, the average value of coal at 50~150˚C remains relatively stable, and the elastic modulus in the temperature range of 150~200˚C shows an increasing trend, and the average value increases from 2.55 to 3.47. The range is 36.39%. After reaching the peak value at 200˚C, the average elastic modulus decreased by 72.62% between 200 and 300˚C. When the oxidation temperature is higher than 200˚C, the internal cohesion of the coal is obviously lost, the stress value decreases greatly, the strain value increases, the brittle-ductile transition occurs, and the elastic modulus decreases. This can be explained by the moisture inside the coal being dried, causing

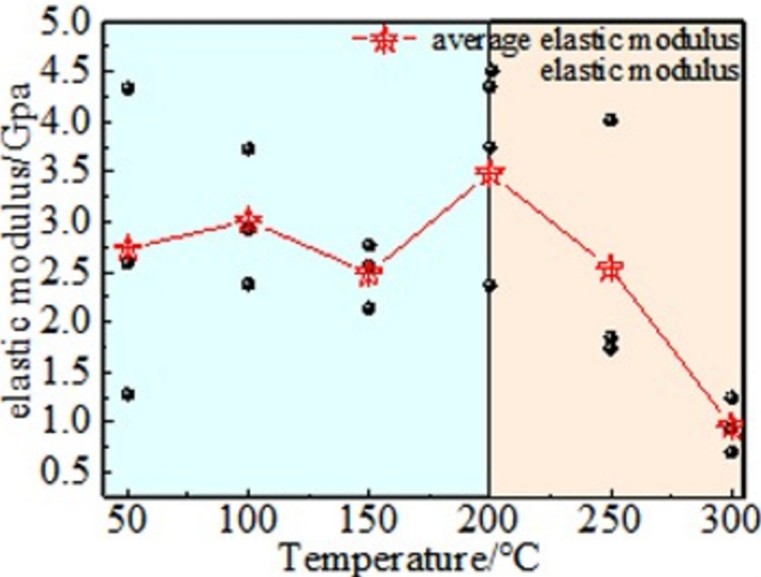

**Fig 17. Elastic modulus.**

the pores to close, and the strength of the coal structure is thus strengthened [33]. When the oxidation temperature is higher than 200°C, the internal cohesive force of the coal is lost, and the stress value is significantly reduced. At the same time, the strain value increases, and brittle-ductile transition occurs, resulting in a significant decrease in the elastic modulus.

## 3.4 Macroscopic destruction of oxidized coal after being loaded

When the temperature was between 50 and 200°C, the pre-peak morphology was jagged, as shown in Figs 18–23. As the load increased, the force structure of the coal sample gradually changed. Although tensile failure occurred locally, the whole remained intact. During the deformation process, some elastic strain energy accumulated, which caused the coal sample to undergo tensile shear failure. When the coal sample underwent fracturing, 3–5 fracture

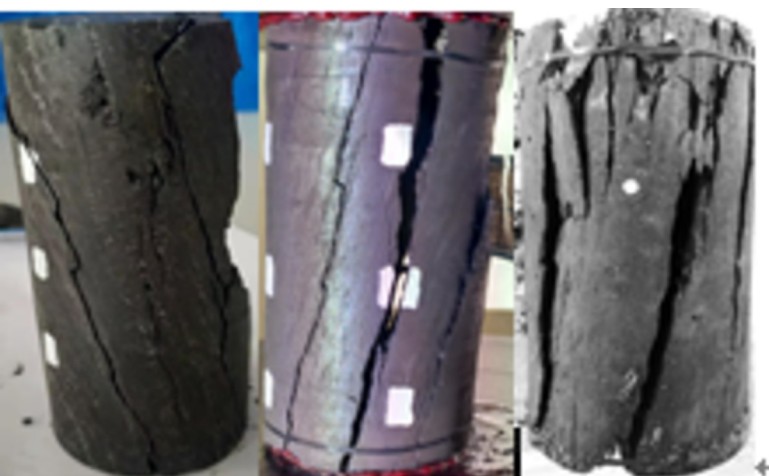

**Fig 18. 50°C.**

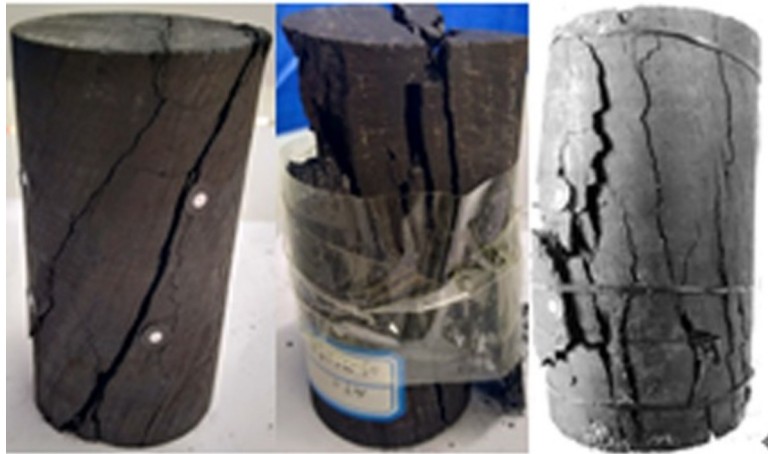

**Fig 19. 100°C.**

surfaces appeared along its axial direction. The specimen was smooth, and the distribution of minor cracks was reduced. Therefore, the energy release was relatively concentrated, a small quantity of semi-penetrating cracks appeared on the surface of each part, and the lines were mostly distributed along the axial direction. Owing to the large elastic modulus, density and strength of coal, the expansion and development of internal microscopic cracks were restricted. Most of the micro-cracks could not continue to develop, but a small part of the micro-cracks continued growing to form perforated macro-micro-cracks, showing brittle failure characteristics. Eventually, the number of primary cracks was reduced, and the secondary cracks disappeared. The curved cracks were gradually replaced by nearly straight cracks, several shear crack bands connected to each other only at the two ends of the specimen, and the crack network structure disappeared. As the treatment temperature was lower, the number of cracks gradually decreased and became more regular when the temperature was between 200 and 300°C. Under the action of external energy, the main crack in the coal sample was obviously affected by the spatial position of the coal microstructure, and the main crack propagation paths increased. Microscopic cracks appeared in multiple areas concentrated in the microstructure of coal, and the shape of the cracks was extremely irregular. The structural

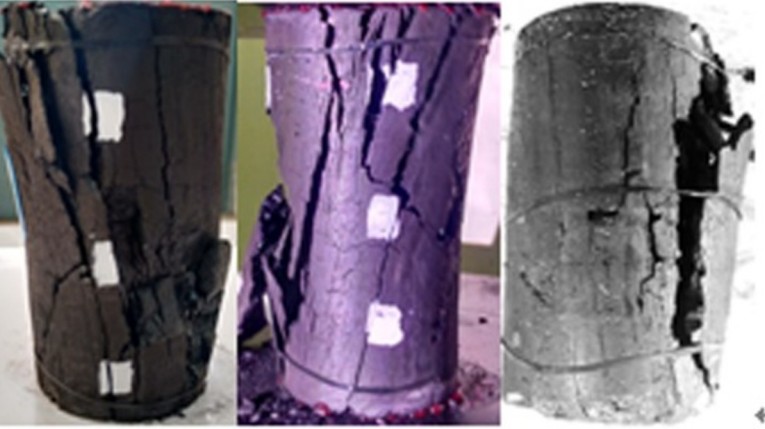

**Fig 20. 150°C.**

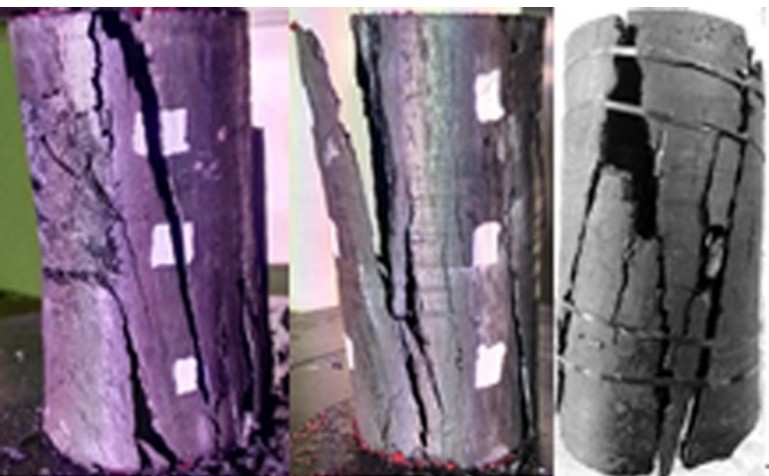

**Fig 21. 200˚C.**

performance exhibited a cross-distribution approximately parallel to the axial direction. The newly formed small cracks merged with the main internal cracks in the coal sample to form a fracture network until failure occurred. There were multiple split surfaces in the coal sample along the axial direction, and many pieces of broken coal were ejected. As a result, there was more damaged before the coal sample was not stressed, showing a large number of clearly visible cracks. As the load was gradually added until the failure process, the cracks in the coal sample were staggered and penetrated, and the degree of fracture increased. Therefore, when the temperature exceeded 200˚C, the mechanical properties of the coal body were expressed as a decrease in hardness and homogeneity. Cleats were relatively developed, the mode of deformation failure was tension-tension failure, and most of the external force was transformed into the dissipated energy of plastic deformation. However, a relatively small amount of elastic strain energy was stored in the coal, and the skeleton structure of the coal body continued to undergo plastic failure. With an increase in the load, the fracture surface of the coal sample increased, the ductile deformation augmented, the energy dissipation increased, and the energy release at the time of failure became smaller.

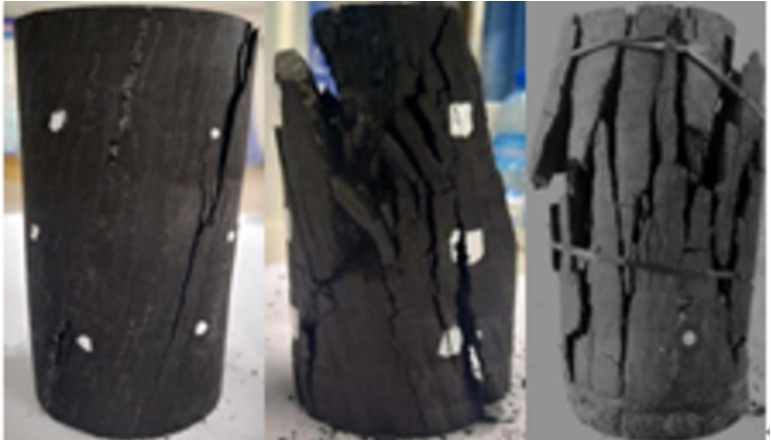

**Fig 22. 250˚C.**

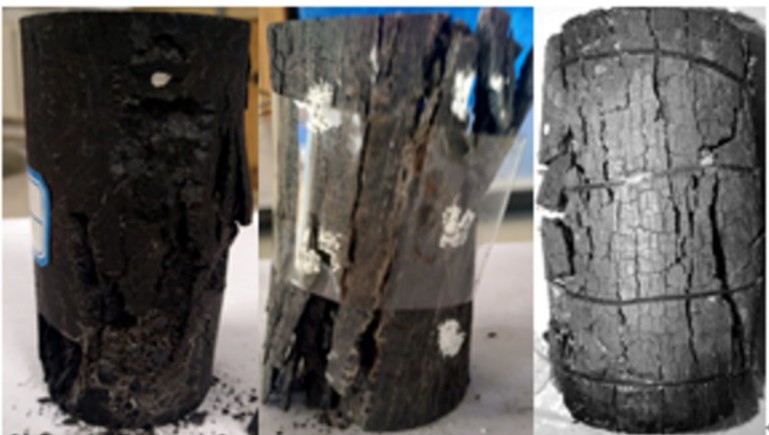

**Fig 23. 300˚C.**

## 3.5 Energy analysis during axial loading of oxidized coal

**3.5.1 Relationship between force and wave velocity.** Figs 24–29 shows the correlation between the wave velocity of non-metallic ultrasonic waves and the externally applied stress under the action of the external load of high-temperature oxidation. The overall change trend of the wave velocity curve is consistent with the stress change. The denser the coal skeleton is, the faster the ultrasonic wave velocity and the better the elastic properties. It shows that the coal sample is broken into macroscopic cracks under the axial load, which leads to the instability and failure of the coal sample in a dynamic evolution process. During the loading process, the coal sample continuously exchanges energy with the testing machine, and the energy conversion causes a dispersion phenomenon inside the coal sample, which continuously changes. The specific manifestations are the appearance and propagation of micro-cracks, interconnection between the cracks, and structural reorganization. The small oscillations in the wave velocity curve indicate that the factors that affect the acoustic wave velocity of the coal are not

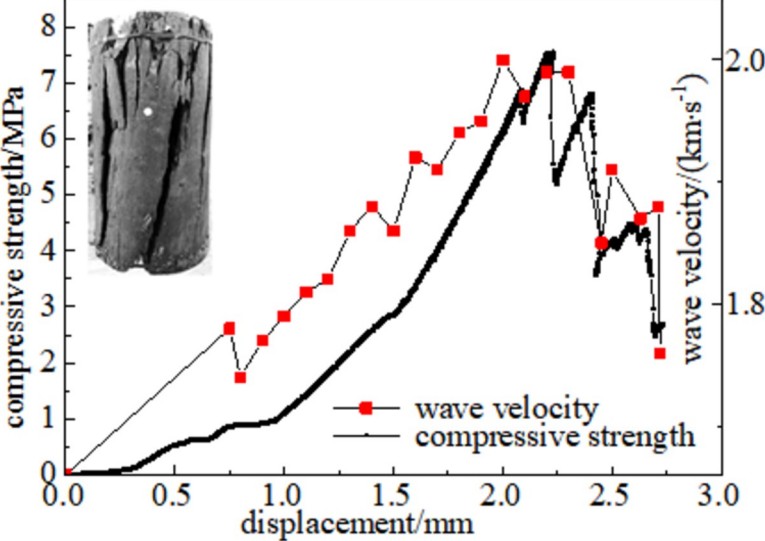

**Fig 24. 50˚C.**

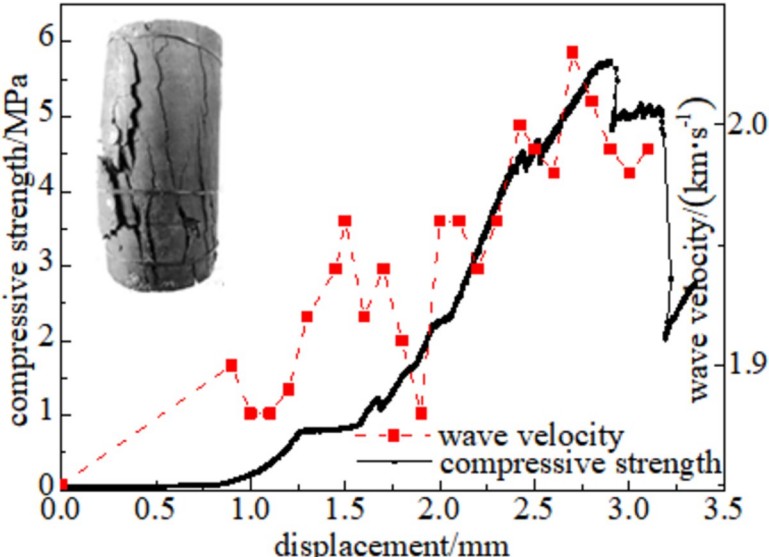

**Fig 25. 100˚C.**

only dependent on the stress, but also related to the number of pores in the coal sample, opening of the gap, and spatial orientation of the crack direction [34].

**3.5.2 Comparison of crack evolution law under normal and high temperature.** The internal structure and fissure degree of the coal sample changed significantly after the coal sample underwent oxidation at different temperatures. To study the crack evolution law under high-temperature treatment, it was observed that the degree of damage of the coal samples became increasingly serious with the increase in temperature. The oxidized coal sample was axially loaded, and the point cloud of the coal sample with different displacements during the fracturing process was collected. The evolution of coal sample cracks can be determined based

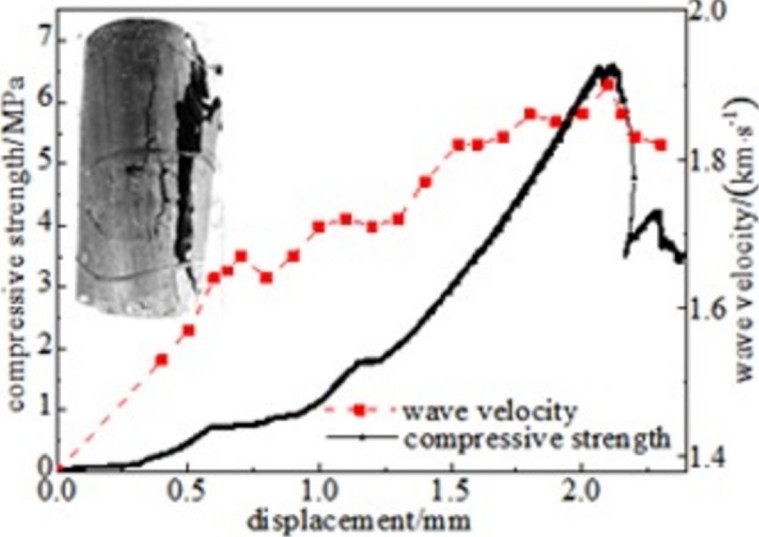

**Fig 26. 150˚C.**

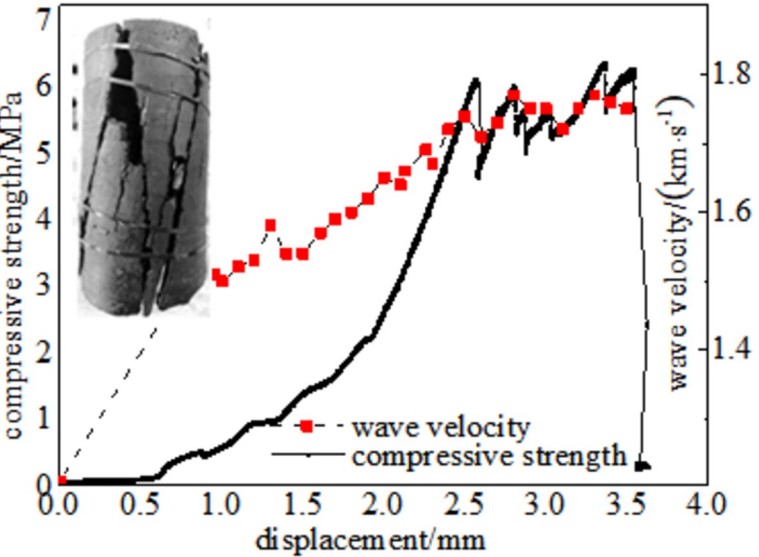

**Fig 27. 200˚C.**

on the collected point cloud of the coal sample. Samples MY19 at 250˚C and MY21 at 50˚C were selected for analysis.

As shown in Fig 30 the coal sample cracks began to appear at points 1 and 2, and then with the axial stress loading, cracks 2 and 3 began to expand until the two cracks penetrated. During the expansion process, the crack width gradually increased, the crack spacing gradually widened, and the crack presented a bifurcated shape. As the axial pressure increased, the bifurcated width formed by the penetration of cracks 2 and 3 gradually widened, and the length gradually increased. The propagating crack is transformed into severe damage caused by local destruction. Meanwhile, four cracks began to occur, and the length of the cracks increased. As the pressure increased, the crack width increased. When reaching the full destruction stage,

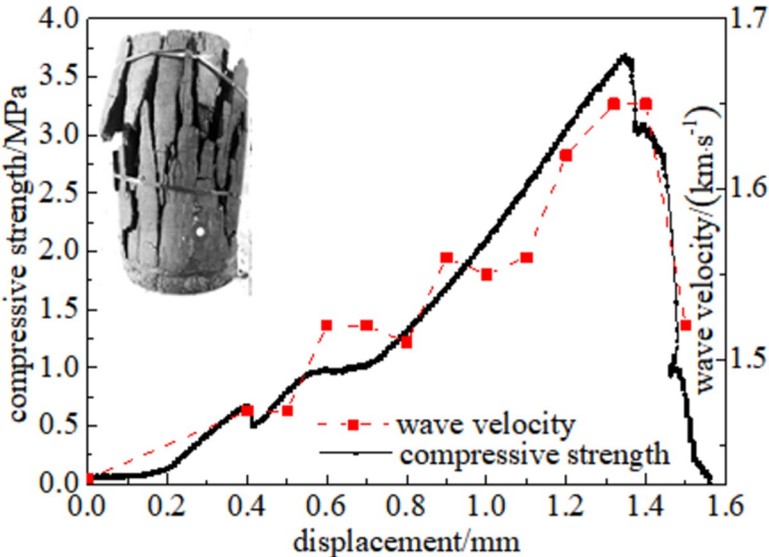

**Fig 28. 250˚C.**

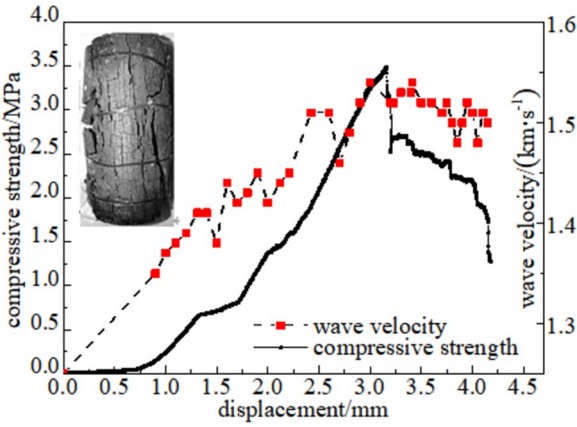

**Fig 29. 300˚C.**

the large cracks formed by the penetration of cracks 2 and 3 were broken, and crack 3 continued to expand. Damage occurred mainly along the two cracks. However, the damage state is not as large as that of the high-temperature coal sample, and the number of cracks formed and the total circumference of the cracks were long.

It can be observed from Fig 31 that the coal body cracks gradually increased as the servo testing machine loaded the coal sample. Coal sample cracks appeared first at A, and cracks B and C began to germinate and extend from A to both sides. As the load increased, the nucleation area at A enlarged, the lengths of cracks B and C increased, and crack D appeared. As the axial stress increased, the lengths of the B and C cracks increased significantly. B-D cracks began to penetrate, and the widths of the B, C, and D cracks augmented. Moreover, the E crack initiation method was consistent with the formation process, and the A, B, and C cracks formed a series structure with other newly formed cracks. Both the loading strength and the main crack width were in a positive phase. Many small cracks sprouted along both sides of the main crack, forming a dendritic structure. As the length of the crack gradually increased, the crack continued to expand. After reaching the peak failure threshold of the coal sample, a large number of micro-cracks were generated before quickly developing into macro-cracks, and a penetration phenomenon appeared. When reaching the full failure stage, the rate of increase

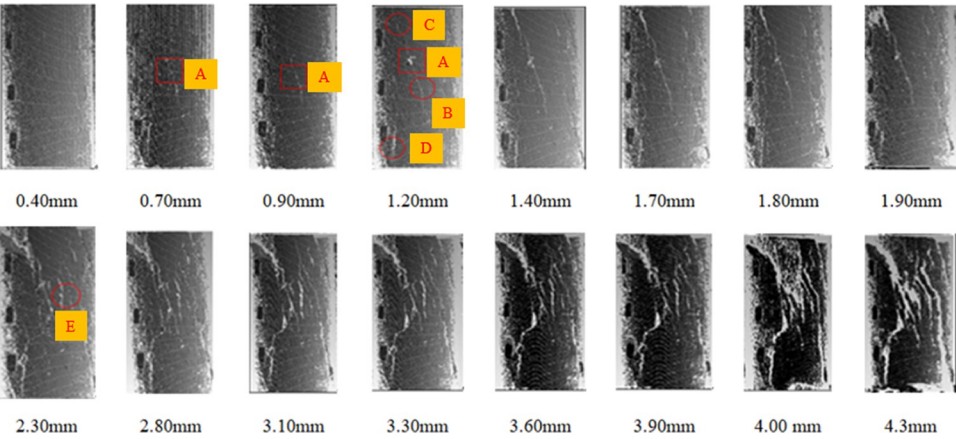

**Fig 30. Displacement failure state of normal temperature coal sample.**

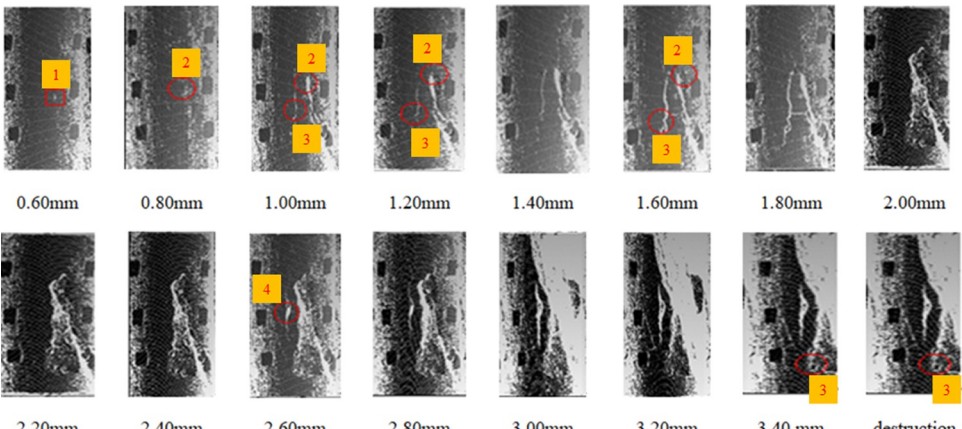

**Fig 31. Displacement failure state of high temperature coal sample.**

in the number and width of cracks increased, and the damage developed from continuous damage to local severe destructive damage.

**3.5.3 Quantitative analysis of oxidized coal energy index.** The coal sample destruction process is accompanied by energy consumption. Owing to the different destruction methods of oxidized coal samples at different temperatures, the corresponding energy loss forms are also different. The failure parameters of coal samples are compressive strength, which is the dynamic failure time, and impacts the energy index, which is analyzed through stress-strain curves of oxidized coal samples at six temperatures. As shown in Fig 32, there was a positive correlation between the wave velocity and compressive strength. In the process of ultrasonic propagation, the elastic wave resistance increases owing to the progressive increase of cracks. This indicates that microstructural changes have a significant impact on mechanical properties. As the temperature increases, compactness gradually decreases. The impact energy index is the ratio of the accumulated deformation energy before the peak of the full stress-strain curve to the loss of deformation energy after the peak. The impact energy index was calculated according to Eq (5) [35].

$$K_\varepsilon = \frac{A_s}{A_x} \tag{5}$$

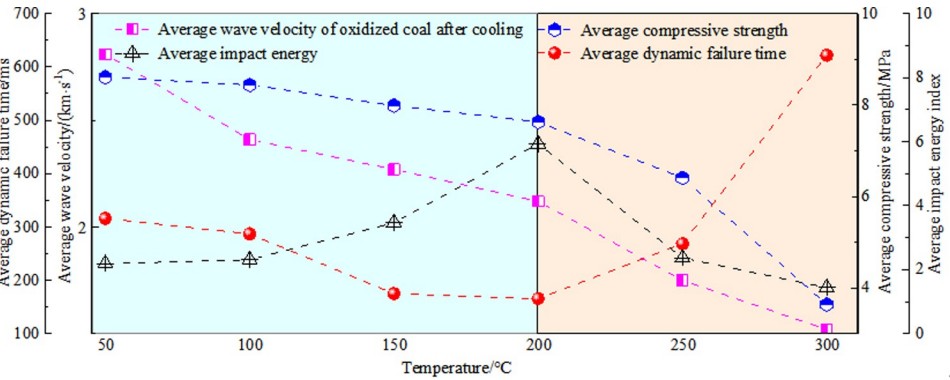

**Fig 32. Analysis of correlation index of coal.**

Fig 32 shows the energy index of the coal samples at different oxidation temperatures. The dynamic failure time of a coal sample ($T_D$) refers to the time from peak strength to complete failure of the coal. The dynamic failure time of coal can characteristic the severity of the release of elastic energy in the case of sudden instability of the coal body. There is a negative correlation between the dynamic failure time ($T_D$) and impact energy index ($K_s$). The average dynamic failure time of the oxidized coal samples showed a decreasing trend, while the impact energy index showed an increasing trend at 50–200˚C. This shows that the moisture inside the coal gradually dried as the temperature increasing, which led to the closure of pores and, cracks and increased the brittleness of the coal. At this stage, the coal is less damaged, the frame bearing capacity is strong, and the storage elasticity before the peak is strong. The coal fractures and destabilizes quickly to release the accumulated elastic energy after reaching the peak. It can be observed that the coal changed from plastic to brittle as time gradually decreased. The dynamic failure time gradually increased, and the impact energy index gradually decreased at 200–300˚C. High temperatures can accelerate the process of coal damage. In the failure state, the coal turned into a loose structure, and the cohesion between the coal particles was lost. In addition, most of the work done by the external force was transformed into plastic deformation dissipated energy. The elastic strain energy stored in the coal was relatively small, and brittle-ductile transition occurred after the peak coal fractures and unstable coal.

## 4 Conclusion

(1) Temperature has a negative effect on the uniaxial compressive strength of coal, and the rate of decrease is slow at 50–200˚C. Later, the rate of decline accelerates, and the high temperature causes the coal sample to oxidize, and increase the porosity. The microscopic damage is visible due to changes in the pore structure and degree of cementation. Moreover, the coal integrity is reduced, and the coal-bearing capacity is weakened.

(2) With the gradual increase in the oxidation temperature, the evolution of coal sample cracks becomes increasingly complicated. The specific manifestations are instantaneous rapid rupture and delayed rupture. The coal sample exhibits obvious ductility characteristics when the pre-peak strain gradually increases.

(3) There is a significant effect of temperature on the Poisson's ratio and elastic modulus of coal. At 50–200˚C, the variation in Poisson's ratio and elastic modulus can be estimated and is based on the temperature. However, when the temperature is higher than 200˚C, both of them will decrease rapidly, and the internal structure will be destroyed.

(4) Wave velocity has an obvious effect on the compressive strength of coal. Before 200˚C, the average dynamic time of the oxidized coal samples decreases, and the impact energy index increases. There is a transition from plasticity to brittleness. After that, the phenomena of dynamic time and energy index are opposite. The fracture surface of the coal sample increases, and the ductile failure characteristics are outstanding. The complexity of network fractures is an essential factor for the brittle-ductile transition in coal.

## Author Contributions

**Conceptualization:** Xiaoqi Wang, Heng Ma, Xiaohan Qi.

**Data curation:** Xiaoqi Wang, Heng Ma, Ke Gao, Xuesong Yang.

**Formal analysis:** Heng Ma.

**Funding acquisition:** Ke Gao.

**Validation:** Shengnan Li.

**Visualization:** Xiaoqi Wang.

**Writing – original draft:** Xiaoqi Wang.

**Writing – review & editing:** Xiaoqi Wang, Heng Ma.

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
