## [Decision Letter · Decision Letter 0]

19 Dec 2021

PONE-D-21-37755Experimental Study on the Effect of High-Temperature Oxidation Coal Mechanical CharacteristicsPLOS ONE

Dear Dr. QI,

Thank you for submitting your manuscript to PLOS ONE. After careful consideration, we feel that it has merit but does not fully meet PLOS ONE’s publication criteria as it currently stands. Therefore, we invite you to submit a revised version of the manuscript that addresses the points raised during the review process.

We look forward to receiving your revised manuscript.

Kind regards,

Marko Čanađija

Academic Editor

PLOS ONE

Journal Requirements:

"This project was funded by the Natural Science Foundation of China (No. 52074148)."

"YES - Specify the role(s) played.a"

"NO authors have competing interests"

7. PLOS requires an ORCID iD for the corresponding author in Editorial Manager on papers submitted after December 6th, 2016. Please ensure that you have an ORCID iD and that it is validated in Editorial Manager. To do this, go to ‘Update my Information’ (in the upper left-hand corner of the main menu), and click on the Fetch/Validate link next to the ORCID field. This will take you to the ORCID site and allow you to create a new iD or authenticate a pre-existing iD in Editorial Manager. Please see the following video for instructions on linking an ORCID iD to your Editorial Manager account: https://www.youtube.com/watch?v=_xcclfuvtxQ

8. We note that Figure 2 in your submission contain [map/satellite] images which may be copyrighted. All PLOS content is published under the Creative Commons Attribution License (CC BY 4.0), which means that the manuscript, images, and Supporting Information files will be freely available online, and any third party is permitted to access, download, copy, distribute, and use these materials in any way, even commercially, with proper attribution. For these reasons, we cannot publish previously copyrighted maps or satellite images created using proprietary data, such as Google software (Google Maps, Street View, and Earth). For more information, see our copyright guidelines: http://journals.plos.org/plosone/s/licenses-and-copyright.

Reviewers' comments:

Reviewer's Responses to Questions

**Comments to the Author**

1. Is the manuscript technically sound, and do the data support the conclusions?

Reviewer #1: Yes

Reviewer #2: Yes

2. Has the statistical analysis been performed appropriately and rigorously? 

Reviewer #1: Yes

Reviewer #2: Yes

3. Have the authors made all data underlying the findings in their manuscript fully available?

Reviewer #1: Yes

Reviewer #2: Yes

4. Is the manuscript presented in an intelligible fashion and written in standard English?

Reviewer #1: Yes

Reviewer #2: Yes

5. Review Comments to the Author

Reviewer #1: This paper studies the correlation between thermal damage and mechanical performance of coal after high – temperature oxidation. The research content is original, and it provides a reference for the evaluation of coal seam stability after high temperature oxidation. The content of the paper is wonderful, substantial, and meets the requirements of the journal of ‘PLOS ONE.’ It is likely to be read and cited. Therefore, the paper can be considered for publication after minor revisions. Some comments are provided as follows.

1. The influence of coal oxidation on the mechanical properties of coal was introduced in Section 3.3. When describing Fig.7, only qualitative analysis was performed, and quantitative description was lacking. It is recommended that the author add relevant content.

2. In fact, the calculation formula 1- 5 were not deduced by the authors. Related references should be provided in the paper.

3. The sharpness of some figures in this paper are not enough, such as Fig. 5(a), Fig.10, and Fig.11. Please modify it.

4. There are some minor errors in the format. Such as line 482, Fig.11 should be modified to Fig. 12, and the line 180, the variable of ‘D’ is italicized. Please carefully check and modify.

5. The content of line 551 reference ‘Jiang et al., 2021’ (‘Jiang Y, Zong P, Ming X, Wei H, Qiao Y. High-temperature fast pyrolysis of coal: an applied basic research using thermal gravimetric analyzer and the downer reactor. Energy. 2021;(2):119977.’) has not been cited in the body of the paper, please confirm carefully. Meanwhile, the author needs to check the accuracy of other references.

Reviewer #2: This paper shows that the micro structure of the coal body has a good positive correlation with the coal mechanical characteristics. This paper quantitatively describes the thermal damage of oxidized coal at different temperatures, and the correlation between thermal damage and mechanical properties of high-temperature oxidized coal was explored in this paper. As a result, a reference for the stability evaluation of high-temperature oxidized coal. The paper has some originality and can be considered for publication after minor revisions.1. Section 3.11 "Physical parameter analysis", line 193 on page 9, expressions are inconsistent. For example, the , are block letter, and the , are italics, please express uniformly. 2. The fonts in Fig.1(a), Fig.1(b), Fig.5(a), Fig.9(a), 9(b) are not clear. Please improve the clarity of the figures to ensure the standardization of the paper.3. When describing the coal body being damaged by high temperature, "mesoscopic damage" is used many times in the full paper. But " mesoscopic damage " is not a common vocabulary in this study field, please ensure the accuracy of professional terms.5. Please express the reference format uniformly. For example, the author representation in Reference 10, 25, 26 is inconsistent with others, please double check and modify.

6. PLOS authors have the option to publish the peer review history of their article (what does this mean?). If published, this will include your full peer review and any attached files.

Reviewer #1: No

Reviewer #2: No

---

## [Author Response · Author response to Decision Letter 0]

26 Jan 2022

Dear reviewers,

Thank you for the reviewer’s comments concerning our manuscript entitled “Experimental Study on the Effect of High-Temperature Oxidation Coal Mechanical Characteristics” (ID: PONE-D-21-37755). Those comments are valuable and helpful for revising and the essential guiding significance to our research. We have substantially revised our manuscript after reading the comments produced by the two reviewers. The major corrections in the paper and the responses to the comments are as follows. 

Reply to reviewers：

Reviewer #1:

This paper studies the correlation between thermal damage and mechanical performance of coal after high –temperature oxidation. The research content is original, and it provides a reference for the evaluation of coal seam stability after high temperature oxidation. The content of the paper is wonderful, substantial, and meets the requirements of the journal of ‘PLoS ONE’. It is likely to be read and cited. Therefore, the paper can be considered for publication after minor revisions. Some comments are provided as follows.

1.The influence of coal oxidation on the mechanical properties of coal was introduced in Section 3.3. When describing Fig.7, only qualitative analysis was performed, and quantitative description was lacking. It is recommended that the author add relevant content.

Response: Thank you very much for pointing out this issue in this work.

I have given it serious consideration. Considering the reviewer’s suggestion, we add a detailed introduction to quantitative description and summary of related contents in this work. Section 3.3 (Influence of coal oxidation on the mechanical properties of coal) was carefully revised, and the changes in the revised manuscript-marked copy are as follows. Please refer to pages 15-16 that highlighted in red words.

Please refer to lines 321-327, pages 15-16 that highlighted in red words.

“As shown in Fig. 6(a), after the coal is oxidized at 50 °C higher than the deep well ambient temperature, the compressive strength generally decreases with temperature. After oxidation from 50-200 °C, the uniaxial compressive strength does not change much with the increase of temperature; the average peak strength of coal at 200-300 °C decreases rapidly, and the average peak stress decreases from 8.13MPa at 200 °C to 250 °C 6.4MPa at 300°C and 4.0MPa at 300°C, the reductions reached 21.28% and 50.80%. Compared with the uniaxial compressive strength of 9.92MPa at 50 °C, the average decrease of the compressive strength at 250~300 °C is 35.48%~59.68%.”

Please refer to lines 366-372, pages 17-18 that highlighted in red words.

“Fig 6(c) shows that 200 °C is the threshold temperature for the change of coal elastic modulus, the average value of coal at 50~150 °C remains relatively stable, and the elastic modulus in the temperature range of 150~200 °C shows an increasing trend and the average value increases from 2.55 to 3.47. The range is 36.39%. After reaching the peak value at 200 °C, the average elastic modulus decreased by 72.62% between 200 and 300 °C. When the oxidation temperature is higher than 200 °C, the internal cohesion of the coal is lost, the stress value decreases greatly, the strain value increases, the brittle-ductile transition occurs, and the elastic modulus decreases.”

Thank you again for your meaningful comments!

2. In fact, the calculation formula 1- 5 were not deduced by the authors. Related references should be provided in the paper.

Response: Thank you very much for your comments.

As you mentioned, the formula 1- 5 are not derived by ourselves, but referenced from other papers. We are so sorry for not citing these contents. To increase the rigor of this paper, relevant references have been added to the revised manuscript-marked copy. See the revised manuscript-marked copy for details in line 575-583, page 27, and line 588-589, page 28.

Relevant references:

Qi XH, Ma H, Wang XQ, Zhang ZG, Lv YC. Impacts of thermal shocks on meso-damage and mechanical properties of coal [J]. China Safety Science Journal, 2020, 30 (12):85-92.

Zhang HW, Wan ZJ, Zhou CB, Zhao YX, Wang W, Yang YL, Teng T. High temperature mechanical properties and thermal shock effect of hot dry rock [J]. Journal of Mining Safety Engineering, 2021 ,38(01):138-145.

Wu X, Liu CW. Evaluating the Proneness of Coal Rockburst Based on the Surface Fractal Feature [J]. Chinese Journal of Underground Space and Engineering, 2013, 9(05): 1045-1049.

Tang SC, Feng P, Zhao JC. Uniaxial Mechanical Properties and Failure Mechanism of Rock Specimens Containing Cross Fissures [J]. Chinese Journal of Underground Space and Engineering, 2021, 17(05): 1376-1383 +1407.

Lu ZG, Ju WJ, Gao FQ, Yi K, Sun ZY. Bursting liability index of coal based on nonlinear storage and release characteristics of elastic energy [J]. Chinese Journal of Rock Mechanics and Engineering, 2021, 40 (08):1559-1569.

3. The sharpness of some figures in this paper are not enough, such as Fig. 5(a), Fig.10, and Fig.11. Please modify it.

Response: Thank you very much for your valuable comments.

Our careless for Fig. 5(a), Fig.10, and Fig.11 are not have enough sharpness. We have redrawn these figures, which have been revised and reordered. The details are as follows. See the revised manuscript-marked copy for more information.

Fig. 5(a) Microscopic image of coal thermal damage after oxidation at different temperatures

Fig. 10. Displacement failure state of normal temperature coal sample

Fig. 11. Displacement failure state of high temperature coal sample

4. There are some minor errors in the format. Such as line 482, Fig.11 should be modified to Fig. 12, and the line 180, the variable of ‘D’ is italicized. Please carefully check and modify.

Response: Thank you very much for your valuable comments.

We are very sorry for the typesetting errors in the paper. The above problems have been corrected (See line 167, page 8). In addition, other places have been checked to make sure there are no errors. See the revised manuscript-marked copy for details.

5. The content of line 551 reference ‘Jiang et al., 2021’ (‘Jiang Y, Zong P, Ming X, Wei H, Qiao Y. High-temperature fast pyrolysis of coal: an applied basic research using thermal gravimetric analyzer and the downer reactor. Energy. 2021;(2):119977.’) has not been cited in the body of the paper, please confirm carefully. Meanwhile, the author needs to check the accuracy of other references.

Response: Thank you for your serious consideration and valuable suggestions on the reference.

However, I found that the content of Reference 11 ('' Jiang Y, Zong P, Ming X, Wei H, Qiao Y. High-temperature fast pyrolysis of coal: an applied basic research using the thermal gravimetric analyzer and the downer reactor. Energy. 2021;(2):119977.'') has been cited in Section Introduction (line 66, page 3) after careful checking.

Reviewer #2:

This paper shows that the micro structure of the coal body has a good positive correlation with the coal mechanical characteristics. This paper quantitatively describes the thermal damage of oxidized coal at different temperatures, and the correlation between thermal damage and mechanical properties of high – temperature oxidized coal was explored in this paper. As a result, a reference for the stability evaluation of high – temperature oxidized coal. The paper has some originality and can be considered for publication after minor revisions.

1. Section 3.11 "Physical parameter analysis", line 193 on page 9, expressions are inconsistent. For example, the , are block letter, and the , are italics, please express uniformly. 

Response: Thank you for your comments.

We have unified the expression and carefully confirmed the accurate expression of other variables in the paper. The following content is marked. Please refer to line 179, page 9 highlighted in red words.

“Second, the coefficients of thermal expansion and modulus of elasticity of the two types of substances are, , and , respectively.”

Thank you again for your comments!

2. The fonts in Fig.1(a), Fig.1(b), Fig.5(a), Fig.9(a), 9(b) are not clear. Please improve the clarity of the figures to ensure the standardization of the paper.

Response: Thank you for your comments on the figures.

We have redrawn these figures, which have been revised and reordered. The details are as follows. See the revised manuscript-marked copy for more information.

 (a)Protection layer mining

(b)Floor gas drainage

Fig.1. Coal oxidation conditions

(a)50℃

(b)100℃

Fig. 10. Relationship between coal compressive strength and sound wave

3. When describing the coal body being damaged by high temperature, "mesoscopic damage" is used many times in the full paper. But " mesoscopic damage " is not a common vocabulary in this study field, please ensure the accuracy of professional terms.

Response: Thank you very much for pointing out this issue in this work. 

I have considered it and read a lot of references. It found that “meso damage” is the professional term used to describe the coal body being damaged by high temperature. A total of 8 in the full paper has been revised. Please refer to the highlighted in red words of the revised manuscript-marked copy.

5. Please express the reference format uniformly. For example, the author representation in Reference 10, 25, 26 is inconsistent with others, please double check and modify. 

Response: Thank you for your consideration and valuable suggestions on the reference.

We have modified and checked the reference format to satisfy PLoS ONE journal publication requirements. Revised references are highlighted with red words in the revised manuscript-marked copy.

Relevant references:

Song J, Deng J, Zhao J, Zhang Y, Wang C, Shu C-M. Critical particle size analysis of gas emission under high-temperature oxidation of weathered coal. Energy. 2021;214.

Yu X, Li G, Chen Z. Uniaxial compressive strength changes of tight sandstone during heating process. Science Technology and Engineering. 2019;019(032):133-8.

Zhang L, Xi L. Study on the uniaxial compression mechanical properties of soft rock after high temperature. Geological Hazards and Environmental Protection. 2019;04(3):583-8.

---

## [Editor Report · Decision Letter 1]

2 Feb 2022

Experimental Study on the Effect of High - Temperature Oxidation Coal Mechanical Characteristics

PONE-D-21-37755R1

Dear Dr. QI,

We’re pleased to inform you that your manuscript has been judged scientifically suitable for publication and will be formally accepted for publication once it meets all outstanding technical requirements.

Kind regards,

Marko Čanađija

Academic Editor

PLOS ONE
---

## [Editor Report · Acceptance letter]

8 Feb 2022

PONE-D-21-37755R1 

Experimental Study on the Effect of High-Temperature Oxidation Coal Mechanical Characteristics 

Dear Dr. Qi:

I'm pleased to inform you that your manuscript has been deemed suitable for publication in PLOS ONE. Congratulations! Your manuscript is now with our production department. 

Kind regards, 

on behalf of

Dr. Marko Čanađija 

Academic Editor

PLOS ONE